# Exploring the Use of Washington Group Questions to Identify People with Clinical Impairments Who Need Services including Assistive Products: Results from Five Population-Based Surveys

**DOI:** 10.3390/ijerph19074304

**Published:** 2022-04-03

**Authors:** Dorothy Boggs, Hannah Kuper, Islay Mactaggart, Tess Bright, GVS Murthy, Abba Hydara, Ian McCormick, Natalia Tamblay, Matias L. Alvarez, Oluwarantimi Atijosan-Ayodele, Hisem Yonso, Allen Foster, Sarah Polack

**Affiliations:** 1International Centre for Evidence in Disability, London School of Hygiene & Tropical Medicine, London WC1E 7HT, UK; hannah.kuper@lshtm.ac.uk (H.K.); islay.mactaggart@lshtm.ac.uk (I.M.); brightt@unimelb.edu.au (T.B.); sarah.polack@lshtm.ac.uk (S.P.); 2International Centre for Eye Health, London School of Hygiene & Tropical Medicine, London WC1E 7HT, UK; murthy.gvs@iiphh.org (G.M.); ian.mccormick@lshtm.ac.uk (I.M.); allen.foster@lshtm.ac.uk (A.F.); 3Indian Institute of Public Health, Hyderabad 122002, India; 4Sheikh Zayed Regional Eye Care Centre, Banjul P.O. Box 650, The Gambia; ahydara@gmail.com; 5Department of Otorhinolaryngology, Faculty of Medicine, Universidad de Chile, Santiago 7790680, Chile; ntamblay@gmail.com; 6Department of Otorhinolaryngology, Faculty of Medicine, Pontificia Universidad Católica de Chile, Santiago 8330077, Chile; mlalvarez@uc.cl; 7Maidstone and Tunbridge Wells NHS Trust, Maidstone ME16 9QQ, UK; rantimi@gmail.com; 8Relief International, Istanbul 34087, Turkey; hisem.yonso@ri.org

**Keywords:** surveys, impairment, functioning, screening, rehabilitation, assistive products, Cameroon, Chile, India, The Gambia, Turkey

## Abstract

This study analyses the use of the self-reported Washington Group (WG) question sets as a first stage screening to identify people with clinical impairments, service and assistive product (AP) referral needs using different cut-off levels in four functional domains (vision, hearing, mobility and cognition). Secondary data analysis was undertaken using population-based survey data from five countries, including one national survey (The Gambia) and four regional/district surveys (Cameroon, Chile, India and Turkey). In total 19,951 participants were sampled (range 538–9188 in individual studies). The WG question sets on functioning were completed for all participants alongside clinical impairment assessments/questionnaires. Using the WG “some/worse difficulty” cut-off identified people with mild/worse impairments with variable sensitivity (44–79%) and specificity (73–92%) in three of the domains. At least 64% and 60% of people with mild/worse impairments who required referral for surgical/medical and rehabilitation/AP services, respectively, self-reported “some/worse difficulty”, and much fewer reported “a lot/worse difficulty.” For moderate/worse impairment, both screening cut-offs improved identification of service/AP need, but a smaller proportion of people with need were identified. In conclusion, WG questions could be used as a first-stage screening option to identify people with impairment and referral needs, but only with moderate sensitivity and specificity.

## 1. Introduction

In global health, alongside mortality and morbidity data, there is an increasing emphasis on addressing a third health indicator, “functioning” [1,2]. The International Classification of Functioning, Disability and Health (ICF) defines functioning as an umbrella term for body functions, structures, activities and participation; it denotes the interaction between an individual (with a health condition) and the environmental and personal context in which they live [3]. The World Health Organization (WHO) estimates that at least 2.4 billion people have difficulties functioning with a need for rehabilitation [4], and more than 1 billion people need assistive technology (AT) with this expected to increase to 2 billion by 2050 [5]. These global estimates are based on assumptions and extrapolations from large population-based impairment datasets, such as the global burden of disease data (GBD).

Diverse groups, including people with disabilities, older people and people with chronic conditions, living in different socioeconomic settings, could benefit from well-planned and resourced services for rehabilitation and assistive products (AP) [6,7]. However, there is currently a lack of data to plan these services. Consequently, there is a need to develop and refine existing survey methodologies to provide population-based data at national and local level on functioning and the need for rehabilitation services and assistive products (APs). These data are particularly needed in low- and middle-income country settings where access to both rehabilitation and APs are often limited, so service availability can be improved and barriers can be addressed [8].

Functioning can be measured through different methodologies including “self-reporting” and clinical assessment [2,9,10]. Self-reporting methodologies are questionnaire based, low cost and rapid to administer. For example, the three main Washington Group (WG) question sets (Short Set, Extended Set and Child Functioning Module) are validated and used widely in population-based disability surveys. They provide self-assessment (or proxy reported) data on components of functioning, predominantly reporting activity limitations across functional domains, including vision, hearing, mobility and cognition, using a four-part scaled response of: no difficulty, some difficulty, a lot of difficulty or cannot do at all [11]. Clinical assessment methodologies in population-based surveys are typically impairment focused and rely on clinicians to diagnose impairments and assess the need for medical, surgical and rehabilitation services, including APs. Clinical impairment assessment is time consuming, requires trained clinicians, is expensive, and often focuses on just one ICF component (impairment), lacking a broader assessment of the individual’s functioning. However, this method provides a more accurate assessment of the need for rehabilitation and AP than self-report alone [9,12,13]. Recently, more “rapid” population-based clinical impairment assessment survey methodologies integrating mobile health technology have been developed which overcome some of the disadvantages of traditional clinician-led measures; for example, the Rapid Assessment of Hearing Loss (RAHL) [14] uses the mobile audiometry tool HearTest together with a clinical examination of the ear [15].

A combination of both self-report and broader clinical assessment is required to obtain more holistic estimates of functioning across multiple domains [2,12,13]. Previous research suggested that a first stage screening using the WG questions, followed by clinical impairment assessment on people who reported “some difficulty” or worse in the corresponding functional domain, would identify the majority of people with impairments and activity limitations [10]. However, evidence is lacking on the appropriateness of this two-stage approach within specific functional domains and on the validity of using different cut-off levels for both self-reported difficulty and clinical impairment. Studies in Fiji with school-age children found that using a cut-off of “some difficulty” in at least one WG domain with accompanying clinical assessments could be used to identify children who require services and learning support; however, the study noted there was widespread variability of identification within impairment levels [16,17,18,19,20]. Evidence from all age groups is lacking, and it is also unclear to what extent this two-stage approach would be able to identify people with specific referral needs (e.g., surgery, rehabilitation and AP). This information is critical to inform service and policy planning and the development of future multi-domain population-based survey tools.

This study aims to address these questions through secondary analysis of datasets from five population-based surveys across the four functional domains of vision, hearing, mobility and cognition that used both WG questions and clinical impairment assessment to examine this two-stage approach.

Specifically, we assess:(1)the sensitivity and specificity of the WG questions (at different cut-off levels) to identify people with clinically assessed impairments (vision, hearing, mobility and cognitive domains);(2)the proportion of people identified by the WG questions (at different cut-off levels) who are in need of surgical/medical and/or rehabilitation/AP services.

## 2. Methods

### 2.1. Population-Based Surveys

This study uses data from five cross-sectional surveys undertaken in Cameroon, Chile, India, The Gambia and Turkey between 2013 and 2020 (Table 1) [12,13,21,22,23,24,25,26,27,28,29,30,31,32]. Four were regional/district surveys (Cameroon, Chile, India and Turkey) and one was a national survey (The Gambia). All surveys used two-stage cluster random sampling. Three surveys included children aged 2 years and over (Cameroon, India and Turkey); The Gambia survey included adults aged 35 years and over; and the Chile survey included adults aged 50 years and over.

### 2.2. Definitions

See Box 1 for definitions of vision, hearing, mobility, cognitive and WG terms used in this paper.

Box 1Definition of vision, hearing, mobility, cognitive and Washington Group terms used in this paper.
**Domain**

**Definition**

**Term Used in This Paper**

**Vision**


Mild or worse vision impairmentPresenting (i.e., with correction, if available) VA < 6/12 in the better eyeMild+ VIModerate or worse vision impairmentPresenting (i.e., with correction, if available) VA < 6/18 in the better eyeModerate+ VINear vision impairmentCannot see N8 at 40cms with correction, if availableNear VI
**Hearing**


Mild or worse hearing impairment>25 dB PTA in the better earMild+ HIModerate or worse hearing impairment>40 dB PTA in the better earModerate+ HI
**Mobility**


Mild or worse musculoskeletalimpairmentAccording to RAM criteria, any participant screening positive underwent clinician assessment to determine presence, severity (mild/moderate/severe) and cause/diagnosis of MSIMild+ MSIModerate or worse musculoskeletal impairmentAccording to RAM criteria, any participant screening positive underwent clinician assessment to determine presence, severity (moderate/severe) and cause/diagnosis of MSIModerate+ MSI
**Cognition**


Cognitive impairment<13 points in the SCh-MMSEMild+ cognitive impairment
**Washington Group Questions (with or without assistive product)**
Some difficulty or worse
Some+ difficultyA lot of difficulty or worse
A lot+ difficulty Abbreviations: VA = visual acuity; VI = vision impairment; PTA = pure tone audiometry; HI = hearing impairment; RAM = Rapid Assessment of Musculoskeletal Impairment; MSI = musculoskeletal impairment; SCh-MMSE = Short Chilean Mini Mental State Examination (SCh-MMSE).

### 2.3. Washington Group Question Sets for Four Domains

Participants (or representatives if unable to self-report) were asked a series of questions from the WG sets. Adults > 17 years were asked either the WG Short Set, the Short Set-Enhanced or the Extended Set on functioning, and children aged 2–17 years were asked the Child Functioning Module [11]. See Box 2 for the WG questions analysed in this paper. The question sets also ask about use of glasses, hearing aids and mobility APs/assistance.

For the purpose of our analyses, we used the screening cut-offs of “some difficulty or worse” (herein referred to as some+) and “a lot of difficulty or worse” (a lot+) either without or with the corresponding AP (if the participant used them) in each of the four functional domains.

Box 2Relevant Washington Group functioning question sets for vision, hearing and mobility functional domains [11].**I. Short Set (SS), Short Set-Enhanced (SS-E) and Extended Question (ES) Set on Functioning Questions (>17 years old**) ^A. Vision1. Do you wear glasses? (Yes/No)2. If yes, do you have difficulty seeing even when wearing your glasses?(No difficulty/Some difficulty/A lot of difficulty/Cannot do at all or unable to do)3. If no, do you have difficulty seeing?(No difficulty/Some difficulty/A lot of difficulty/Cannot do at all or unable to do)B. Hearing1. Do you wear a hearing aid? (Yes/No)2. If yes, do you have difficulty hearing even when using a hearing aid? (No difficulty/Some difficulty/A lot of difficulty/Cannot do at all or unable to do3. If no, do you have difficulty hearing? (No difficulty/Some difficulty/A lot of difficulty/Cannot do at all or unable to do)C. Mobility1. * Do you use any equipment or receive help for getting around? (Yes/No)2. * If yes, do you have difficulty walking or climbing steps, even when using your equipment or with help? (No difficulty/Some difficulty/A lot of difficulty/Cannot do at all or unable to do)3. Do you have difficulty walking or climbing steps? (No difficulty/Some difficulty/A lot of difficulty/Cannot do at all or unable to do)D. Cognition1. Do you have difficulty remembering or concentrating? (No difficulty/Some difficulty/A lot of difficulty/Cannot do at all or unable to do)**II. Child Functioning Module (5–17 years old) and Child Functioning Module (2–4 years old) ^**A. Vision (same as above applied to the child and asked of the carer)B. Hearing (same as above applied to the child and asked of the carer)C. Mobility1. ** Compared with children of the same age, does [name] have difficulty walking?2. Does (name) use any equipment or receive assistance for walking?3. Without his/her equipment or assistance, does [name] have difficulty walking 100 m on level ground? That would be about the length of one football field.4. Without his/her equipment or assistance, does [name] have difficulty walking 500 m on level ground? That would be about the length of five football fields.5. With his/her equipment or assistance, does [name] have difficulty walking 100 m on level ground?6. With his/her equipment or assistance, does [name] have difficulty walking 500 m on level ground?7. Compared with children of the same age, does [name] have difficulty walking 100 m on level ground?8. Compared with children of the same age, does [name] have difficulty walking 500 m on level ground?^ Subsets of full questionnaires; * Questions not asked in Turkey Disability and Mental Health Survey; ** Only question asked in Cameroon and India Disability Surveys, all other questions were asked in Turkey Disability and Mental Health Survey only.

### 2.4. Clinical Assessment and Questionnaires

#### 2.4.1. Vision

Distance vision: 

Presenting visual acuity (VA) (i.e., with correction, if available) was assessed using a tumbling E single optotype, on cards in India and Cameroon and on Peek Acuity mobile application in The Gambia [33]. In India and The Gambia mild or worse vision impairment (VI) was defined as presenting VA < 6/12 in the better eye; and in India, The Gambia and Cameroon, moderate or worse VI was defined as presenting VA < 6/18 in the better eye [34]. Pinhole vision was assessed for all participants with VI to identify individuals with uncorrected refractive error (URE), and in The Gambia a refraction was performed to record best corrected visual acuity (BCVA). In all three countries (Cameroon, India and The Gambia), participants with VI were examined by a trained eye care worker to determine the cause using the WHO protocol for the condition that is “easiest to treat” [35]. Those identified with URE were reported as needing distance glasses.

Near vision:

Presenting (i.e., with near correction if available) binocular near vision was assessed in The Gambia survey only. A binary outcome of can or cannot see N8 at 40 cm (correctly identifies 4 out of 5 E optotypes) was recorded. Participants unable to see N8 were re-tested using an age-appropriate correction for near and recorded as needing near glasses (unmet need) or needing a change in prescription of existing near glasses (undermet need).

#### 2.4.2. Hearing

In India and Cameroon, all-age participants were screened using Otoacoustic Emissions (OEA) Testing, and participants aged ≥ 4 years old who failed this underwent Pure Tone Audiometry (PTA) at 0.5, 1, 2 and 4 kHz to assess the presence and severity of hearing loss. The definition of moderate or worse hearing impairment (HI) was a pure tone average (at 0.5, 1, 2 and 4 kHz) of >31 dB for children (4 to 17 years) and >41 dB for adults (≥18 years) in the better ear [36]. Individuals with HI underwent examination by an ENT specialist to assess the cause and likely service needs, including hearing aids. 

In Chile [14], PTA was tested using a mobile-based audiometry system HearTest [37] at 0.5, 1, 2 and 4 kHz in each ear. According to WHO’s definition, mild or worse HI was defined as >25 dB in the better ear. All participants had their ears examined by an ENT resident or consultant.

#### 2.4.3. Mobility

In Cameroon, India and Turkey, participants were asked six validated screening questions from the Rapid Assessment of Musculoskeletal Impairment survey tool (RAM) [38]. Anyone who screened positive underwent a standardised examination by a physiotherapist using the RAM protocol which includes head/neck, upper limb, lower limb/pelvis, trunk and spine assessment. The presence, severity (mild/moderate/severe) and cause/diagnosis of MSI, as well as the need for services and APs including wheelchairs and prosthetics (both upper and lower limb) was recorded. In Turkey an updated version of RAM was used [27].

#### 2.4.4. Cognition

The Mini Mental State Examination (MMSE) is a brief neurophysiological test [39]. A short validated Chilean Mini Mental State Examination (SCh-MMSE) was developed by an audiologist in Chile [40] to include populations with low levels of literacy [41]. This includes six questions selected from the original 11 question version. The six questions evaluate:-spatial and temporal orientation (day, month, year);-short- and long-term memory (3 word retention);-attention (inverse repetition of 5 numbers);-executive capacity (verbal order with 3 steps);-visual constructive capacities (copy of two circles).

Each of the questions has a score, with a possible maximum of 19 points; a total score <13 is considered “suspected cognitive impairment.”

### 2.5. Data Analysis

Stata version 16.0 (StataCorp LP, College Station, TX, USA) was used to manage and analyse the data. The cluster design was accounted for in the analysis using the “svy” command.

To test whether the WG self-reporting questions, as a first-stage screen, are able to identify people assessed as having clinical impairments, we calculated sensitivity, specificity, positive predictive value and negative predictive values, with clinical impairment assessment being the reference.

To understand the extent to which the WG questions identify people who could benefit from referral for a specific intervention, we calculated the proportion of individuals who by clinical impairment assessment were found to need surgical/medical or rehabilitation/AP interventions who self-reported “some+ difficulty” or “a lot+ difficulty” for both mild+ and moderate+ impairment levels. For the purposes of our analyses, we restricted clinically assessed service and AP need to only those participants who responded to the corresponding WG question in each domain. Surgical/medical and rehabilitation/AP service needs, and need for five individual APs classified as “priority APs” by ATScale [42] (distance glasses, near glasses, hearing aids, wheelchairs and prosthetics), were clinically assessed according to cause, diagnosis and severity. Domain-specific details, used for our analysis, are provided below.
-Vision: For participants with vision loss due to cataract, surgical intervention was assigned. For participants with URE, distance glasses were assigned as the intervention. For participants with other causes of visual loss, e.g., glaucoma, both medical and rehabilitation services were recorded, and, for causes with no medical or surgical treatment possibilities, only rehabilitation services were assigned.-Hearing: Following the protocol used in RAHL [14], for participants with hearing loss due to chronic otitis media (dry/wet/possible Cholesteatoma), acute otitis media, otitis media with effusion, otitis externa, impacted wax and foreign body, surgical/medical intervention was assigned. Participants with sensorineural/mixed hearing loss in both ears, or unknown cause, were categorised as needing “referral to audiological rehabilitation services and likely hearing aids”. In Cameroon and India, clinician-assessed hearing aid referrals were used.-Mobility/MSI: According to the RAM [38], surgical/medical and rehabilitation/AP interventions were clinically assessed based upon the examination with specific referral recommendations recorded by the clinician. For example, rehabilitation services included referrals to physiotherapy and environmental modifications, and APs including up to 11 mobility APs, such as wheelchairs, prosthetics, sticks/canes and orthotics.

In each of the functional domains, some participants were assessed to need both medical/surgical interventions and rehabilitation/AP services. Data on intervention need were not available for the cognition domain.

### 2.6. Ethics and Consent

This secondary analysis study received approval from the London School of Hygiene & Tropical Medicine. Each survey received separate approval from the London School of Hygiene & Tropical Medicine and the relevant ethics committees in each study country [10,27,28,30,31]. Written (signed or fingerprinted) informed consent was obtained from all participants or their proxies.

## 3. Results

### 3.1. Overall Survey Results

Table 1 presents the survey details for each country. The sample size ranged from 538–9188 participants with response rates of 47% to 88%. In all five surveys at least half of the study population were female.

### 3.2. WG Questions to Screen for Clinical Impairment

The association between clinical impairment assessment and self-reported difficulty in functioning for each domain is presented in Table 2 with additional analyses in Appendix A.

Across the different impairments and study settings, using the WG category “some or worse” difficulty identified people with clinical impairments with a sensitivity range of 44% to 85%, and a specificity range of 65% to 92%. There was one exception of very low specificity (18%) for mild+ cognitive impairment (Chile). Using the more restrictive “a lot or worse” difficulty consistently, across impairment types and studies, reduced sensitivity (range 9–62%) and improved specificity (range 86–99.7%). “Near VI” was only measured in The Gambia and had low/very low sensitivity (39% and 3%) and high specificity (85% and 99.5%) using both WG cut-offs of some+ and a lot+ difficulty, respectively.

Specific ranges in each domain were as follows. For distance vision, self-reported WG “some+ difficulty” seeing had good/high sensitivity (67–85%) and specificity (77–80%) when compared to clinical VI. Moving to a cut-off of “a lot+ difficulty” increased the sensitivity (99%) but radically reduced the specificity (10–39%). For hearing, “some+ difficulty” hearing had moderate/high sensitivity (44–83%) and good/high specificity (65–92%) when compared to clinical HI. Moving to a cut-off of “a lot+ difficulty” increased the sensitivity (98–99.7%) but reduced the specificity (9–50%). For mobility, some+ difficulty” walking had good/high sensitivity (64–84%) and specificity (76–90%) when compared to clinical MSI. Moving to a cut-off of “a lot+ difficulty” increased the sensitivity (97.6–99.7%) but reduced the specificity (16–62%).

### 3.3. WG Questions to Screen for Service/Intervention Needs

Table 3 shows the proportion of participants with identified clinical impairment who were assessed to need either medical/surgical interventions (e.g., cataract surgery) and/or rehabilitation/AP services (e.g., hearing aids) who were identified by the WG self-reported questions.

Over three-fifths of participants (range 62–96%) with impairments who needed a surgical/medical intervention self-reported “some+ difficulty”, whereas much fewer (range 13–64%) reported “a lot+ difficulty” across the studies. Of those who needed rehabilitation services and/or APs, 60–86% of persons with impairments self-reported “some+ difficulty” and much fewer (5–62%) reported “a lot+ difficulty.”

The detailed results for each country and service/intervention need are shown in Appendix A. Additionally, only 39% of people who were clinically assessed to need functional near vision services reported “some+ difficulty”. 

Across all domains and countries, the overall population-level need for rehabilitation/AP services (2–43%) was approximately equal to or greater than the need for surgical/medical services (2–10%), except for moderate+ VI in India and The Gambia (see Appendix A).

### 3.4. Identification of Persons Needing Specific Assistive Products

The proportion of people who were assessed as having a clinical impairment who needed glasses (distance and near), hearing aids, wheelchairs and protheses that were identified by WG question “some+ difficulty” is presented in Table 4, with all categories presented in Appendix A.

Of people with mild+ VI who were clinically assessed to need distance glasses, 59–76% reported having “some+ difficulty” seeing, and of those with moderate+ VI, it was 60% to 81%. Of those who were clinically assessed to need near glasses, only 34% of people with near VI reported having “some+ difficulty” seeing. Of the people with mild+ HI who were clinically assessed to likely need hearing aids, 60% reported having “some+ difficulty” hearing, and, of those with moderate+ HI, 72% to 81% reported having “some+ difficulty” hearing. Overall in three countries, 14 of 15 (93%) people clinically assessed as needing wheelchairs, and 4 of 5 (80%) of people who needed a prosthesis reported having “some+ difficulty” walking.

## 4. Discussion

### 4.1. Use of Washington Group Questions for Initial Screening in Population-Based Clinical Assessment Surveys

Overall, using the “some or worse” difficulty cut-off for WG questions demonstrates better agreement with the presence of clinical impairments and service/AP referral needs than using “a lot or worse” difficulty. This pattern remained true for both mild+ and moderate+ impairments in each of the three functional domains of vision, hearing and mobility. Use of “some or worse” WG screening cut-off would identify at least 60% of people with mild+ impairment who could potentially benefit from a service/intervention, but with many false negatives. In contrast, using the cut-off “a lot or worse” difficulty would miss the vast majority of people with service needs. Similarly, the proportion of eligible people identified through WG screening increased when using moderate+ impairment threshold, but a smaller proportion of people with need were identified.

Specifically, our study explored Mactaggart et al.’s recommendation to use a WG cut-off of “some difficulty or worse” as first-stage screening followed by clinical impairment assessment in the same functional domain to identify people with disabilities, based upon a moderate+ impairment threshold [10]. Though our overall findings were congruent with the general recommendation to use “some difficulty or worse” cut-off, Mactaggart et al.’s research anticipated that *at least 80%* of people with disabilities would be identified using this method, whereas our study found much fewer people with impairment (44–79%) and people with service/AP needs (60–82%) would be identified using updated recommended mild+ impairment thresholds [2,12,13,14,27,43,44]. Therefore, it appears use of this screening recommendation might not be transferrable to a mild+ impairment threshold.

There are few population-based prevalence studies that allow comparison with our findings to ascertain what might be a recommended “minimum” identification screening threshold. A few hearing impairment studies exist, and one study similarly found a self-report screen identified 80% of people with clinically assessed hearing loss [45]. However, regardless of the threshold, it could be argued that using a two-stage screening might indicate the proportion of “service demand” in a given population. Though literature is limited, the rationale for this statement could be that people who report a difficulty in functioning may be more likely to consider that they need services, and therefore uptake related services, creating a “service demand.” In contrast, people who report having no difficulty may be unlikely to uptake referrals for services. For example, a study in New Zealand found that measuring unserviced health needs through a patient-initiated general practitioner consultation was directly relevant to service planning because the gaps identified reflected clinically indicated services that patients want and need [46]. Similarly, this relationship has been evidenced for mental health, where perceived mental health need has been shown to be predictive of seeking services [47,48]. However, research has also recognised that demand-based health needs planning could increase access and utilisation service gaps and inequities between social groups in populations; therefore, it is recommended that demand-based health needs planning should also be coupled with need-based allocation of resources and a focus on the empowerment of groups who have greater needs [49]. There is a need to further explore this relationship in the context of collecting population-based data to inform service planning.

Our findings were closely aligned with Sprunt et al.’s findings which found variable sensitivity and specificity overall and by impairment severity (none to severe) when exploring the use of the WG question set CFM “some difficulty” as a screening for school-aged children in Fiji with impairments [16,17,18,19,20]. Following their analysis, Sprunt et al. recommended to use the WG first-stage screening of “some difficulty” *in a minimum of one functional domain**,* and that subsequently additional wide-ranging clinical assessments should be administered by the school system in Fiji in order to pick up unidentified and unexpected impairments [16,17,18,19]. Therefore, it is recommended that future research explore this additional analysis as an option for population-based multi-domain survey two-stage screening, whilst parallel research should also consider the feasibility, affordability and acceptability of administration. Further, the Fiji study also specifically highlighted the importance of including environmental factors specific for learning and support needs [19]. Therefore, other potential screening and clinical assessment tools incorporating more environmental factors should be explored.

Our study has shown that the proposed use of the WG “some+ difficulty” as a first stage screening could be a practical and feasible option to reduce the survey duration, cost and response burden compared to conducting multi-domain impairment assessments. However, our findings have also shown that this approach will not capture everyone with impairments and service/AP needs in each domain so it will not be appropriate for surveys that aim to estimate prevalence of impairment and service/AP need. 

### 4.2. Further Gaps in the Survey Measurement Approaches

Our analysis highlighted gaps and recommendations to be considered in the collection of data on functional service needs.

First, rehabilitation/AP needs are often neglected, but this paper highlights that need was at least equal to or higher than the surgical/medical service need across all countries and domains. This further highlights the importance of increased data collection efforts, using robust methodology, to assess need in different settings. 

Second, adjustments to the first stage screening questions might be needed. For example, in the vision domain, there was poor identification of near VI service/AP needs using both “some+ difficulty” and “a lot+ difficulty” WG cut-off levels (39% and 3%, respectively). This may be expected as the specific WG question asked in general about difficulty seeing (see Box 2). Therefore, it is recommended that a specific near vision screening question is included for surveys that intend to assess near VI and the need for services and AP to improve the sensitivity. The WG extended question set provides an optional vision question that asks about a functional activity related to use of near vision—difficulty clearly seeing the picture on a coin—so this should be incorporated at a minimum [11]. In the mobility domain, the extended WG questions for difficulty walking over certain distances and/or climbing stairs were compared to MSI (Box 2). Though walking could be one activity limitation for a person with MSI, other possible activity limitations also assessed in WG questions include self-care, upper body, pain and fatigue. Future analysis should explore whether combinations of these questions, in addition to environmental questions, increase identification of people with MSI and service needs by improving the sensitivity and specificity. Furthermore, consideration should be given as to whether additional first-stage screening options might provide better prevalence estimates of need, such as the validated RAM’s six self-reported first-stage screening questions [38].

Third, the development of a multi-domain modular survey tool would allow flexibility, depending on aim of the data collection and time and resources available. For example, this could include options to (i) include or not include the first stage WG screen in the survey and (ii) select which functional domains to include.

Fourth, this paper used secondary analysis of datasets and the analyses were therefore constrained by data that were collected. For example, in the cognitive domain, there was very low specificity (18%) for mild+ cognitive impairment without service recommendations. Though the SCh-MMSE was contextually developed for low literacy populations which is a strength, using this screening tool as a “gold standard” in our study has limitations. Future studies are needed to explore and compare additional cognitive assessment and screening tools which include assessments of cognitive service/AP needs as well. 

Fifth, when using the WG questions in service/AP need surveys, consideration could be given to ask about the presence of functional difficulties *without* the use of assistance or APs. For example, Danemayer et al.’s systematic review recommended AP indicators of total need and met need, as well as unmet and undermet need for service/AP need, are collected in population-based surveys [50]. To collect these data, a first-stage screening would also need to capture people who are using services/APs who could then undergo impairment assessment. Future research could consider asking participants about reported functional difficulties *without* assistance/APs as well which collects important quality of services data. The WHO Rapid Assessment of Assistive Technology [51,52] uses the WG Short Set with this modification and includes an additional survey section which asks about broader AP use; however, this sole modification generates non-comparable WG functioning data so is not a viable approach.

Finally, when exploring options for the second stage of a two-stage survey to estimate participants’ functional service and AP needs, it is important not to rely on clinical assessments solely measuring impairments since this more ”medical” model of assessment is only estimating one ICF component of functioning. It is key that second-stage assessments integrate broader functioning components when developing survey tools for assessing need, including consideration of environmental factors as highlighted by Sprunt et al. Therefore, alongside clinical impairment assessments, more hybrid clinical assessments measuring broader functional needs should be incorporated through structured observation and demonstration of tasks/activities, in addition to self-reported measures on activities, participation and environmental factors. This integration would ensure enhanced alignment with the ICF’s broader definition of functioning and also would provide more detailed data about specific rehabilitation/AP service and human resource needs for evidence-based health and social policy and planning beyond solely surgical and medical needs.

### 4.3. Strengths and Limitations

The biggest strength is that approximately 20,000 survey participants from five countries are included in these secondary analyses presenting important data on the potential use of WG questions as first-stage screening questions in population-based surveys. We compared two cut-off levels of self-reported WG data to two cut-off levels of clinically assessed impairment data, as well as comparing clinically assessed need for services and select APs with the two methodologies. However, there were limitations. First, the authors acknowledge that the comparison in methodologies is based upon two separate ICF components of functioning—(i) impairment or body structure/function, and (ii) activity limitation—using clinically assessed impairment as the ”gold standard.” It is possible that some of the variation between the methodologies was due to measuring two separate ICF components. Second, our analyses in this paper were limited to “unmet” and “undermet” need comparing only a proportion of “total need” in the two methodologies to identify those who needed services. Therefore, we did not consider those who ”use” services and APs that might actually have/had “met,” “overmet” or ”undermet” clinically assessed service and AP needs. Third, for the vision and hearing domains service groupings, we allocated participants to medical/surgical and/or rehabilitation/AP services, and often both, using a list of possible diagnoses. This retrospective allocation is likely to have over-estimated the need for both types of services given certain clinical diagnoses were assigned to both categories. Additionally, bilateral moderate+ impairment might be used as the referral threshold in some countries, such as in Chile for government financed hearing aids for people > 65 years. Fourth, the WG may not be the best possible screening tool, but it was used in the five surveys because it is widely endorsed and utilised. It could be interesting for future research to compare other self-reported functioning survey tools, such as the WHO’s Disability Assessment Schedule [53] and Brief Model Disability Survey [54], with the WG questions for potential screening questions in multi-domain population-based surveys. Finally, all five surveys were supported by the same research group, the International Centre of Evidence in Disability at the London School of Hygiene & Tropical Medicine, using similar methodologies for clinically assessing vision, hearing and mobility impairments and service/AP need. While this is a strength in terms of comparability of methods, it is also possible that the use of alternate and/or additional methodologies and/or tools incorporating broader functioning components might have provided different results and should be considered in future research.

## 5. Conclusions

This paper explores the use of self-reported WG questions as a first-stage screening in population-based surveys. Our analyses found the WG questions could be used as a first-stage screening *option* to identify people with impairment and referral needs, but only with moderate sensitivity and specificity. If developing a multi-domain hybrid assessment survey tool, it therefore would be important to include options to (i) include or not include the first stage WG screen in the survey and (ii) select which functional domains to include. It is also recommended to explore additional first-stage screening cut-offs and options to provide better prevalence estimates of need, and incorporate assessments for other ICF components, especially personal and environmental factors, for more holistic hybrid methodology assessment of functional needs. Overall, our findings are important for the ongoing development and feasibility testing of population-based survey methodology and survey implementation considerations.

## Figures and Tables

**Table 1 ijerph-19-04304-t001:** Survey participants and clinical impairment assessments in Cameroon, Chile, India, The Gambia and Turkey.

	Cameroon	Chile	India	The Gambia	Turkey
**Overall**					
Place	Fundong Health District (North West)	Province of Santiago	Mahbubnagar District, Telangana	National Survey	Sultanbeyli, District of Istanbul
Year	2013	2019–20	2014	2019	2019
Sample Size	3567	538	3574	9188	3084
Response Rate %	87%	47%	88%	83%	77%
Age Group	2+ years	50+ years	2+ years	35+ years	2+ years
% Female	59%	64%	52%	71%	53%
**Clinical Impairment Assessment Method**					
**Vision Assessment**					
Children and Adults	VA plus clinical examination	-	VA plus clinical examination	Children not assessed; VA and near vision plus clinical examination	-
**Hearing Assessment**					
Children and Adults	OAE, PTA (≥4yo) and clinical examination	Children not assessed; PTA and clinical examination	OAE, PTA (≥4yo) and clinical examination	-	-
**Mobility/MSI Assessment**					
Children and Adults	Clinical mobility assessment	-	Clinical mobility assessment	-	Clinical mobility assessment
**Cognition Assessment**					
Adults Only	-	Standardised questionnaire	-	-	-

Abbreviations: VA = visual acuity; OAE = otoacoustic emissions; PTA = pure tone audiometry.

**Table 2 ijerph-19-04304-t002:** Relationship between self-reported difficulties and clinically assessed impairments by functional domain for vision, hearing, mobility and cognition.

Impairment Severity Levels	N/Total WG Population Assessed	Washington Group Self-Reported Seeing Difficulty Responses							
		Some+ Difficulty				A lot+ Difficulty			
		Sensitivity	Specificity	PPV	NPV	Sensitivity	Specificity	PPV	NPV
**Distance Vision Impairment**									
Cameroon									
Moderate+	82/3314	79%	80%	9%	99%	30%	99%	46%	98%
India									
Mild+	282/3451	79%	80%	26%	98%	18%	99%	58%	93%
Moderate+	119/3451	85%	77%	12%	99%	39%	99%	52%	98%
The Gambia *									
Mild+	1323/9180	67%	79%	35%	94%	10%	99%	70%	87%
Moderate+	998/9180	70%	78%	28%	96%	11%	99%	63%	90%
**Hearing Impairment**									
Cameroon **									
Mild+	271/3005	44%	89%	28%	94%	9%	99.7%	73%	92%
Moderate+	103/3005	66%	88%	16%	99%	20%	99.6%	64%	97%
Chile									
Mild+	225/492	61%	73%	66%	69%	14%	98%	86%	57%
Moderate+	82/492	78%	65%	31%	94%	33%	98%	75%	88%
India									
Mild+	312/3253	60%	92%	44%	96%	25%	99.7%	89%	93%
Moderate+	153/3253	83%	90%	30%	99%	50%	99.6%	85%	98%
**Mobility Impairment**									
Cameroon									
Mild+	423/3308	68%	81%	34%	95%	17%	99%	72%	89%
Moderate+	135/3308	68%	76%	11%	98%	36%	98%	47%	97%
India									
Mild+	694/3439	64%	90%	61%	91%	16%	99.7%	93%	82%
Moderate+	123/3439	84%	81%	14%	99%	62%	98.6%	63%	98.6%
Turkey									
Mild+	365/3014	67%	88%	44%	95%	33%	98.8%	79%	91%
Moderate+	255/3014	70%	86%	32%	97%	33%	97.6%	56%	94%
**Cognitive Impairment**									
Chile									
Mild+	70/534	83%	18%	13%	88%	31%	86%	25%	89%

Abbreviations: WG = Washington Group; PPV = positive predictive value; NPV = negative predictive value. * 8 survey participants were missing WG data. ** Limited to participants ≥ 4 years old with complete PTA; in Cameroon, 11 survey participants were missing WG data.

**Table 3 ijerph-19-04304-t003:** Proportion of participants assessed to have a clinical impairment and need interventions* who were identified as having functional difficulties by Washington Group questions.

	Washington Group Questions					
	No Difficulty		Some+ Difficulty		A Lot+ Difficulty	
Domain	Need Medical/Surgical Intervention	Need Rehab. Services/APs	Need Medical/Surgical Intervention	Need Rehab. Services/APs	Need Medical/Surgical Intervention	Need Rehab. Services/APs
Mild VI < 6/12	18–27%	26–38%	73–82%	62–74%	13–33%	5–8%
Moderate VI < 6/18	15–26%	24–35%	74–85%	65–76%	14–40%	10–33%
Mild HI	34%	40%	66%	60%	20%	13%
Moderate HI	4–38%	18–33%	62–96%	67–82%	12–64%	25–50%
Mobility: Mild MSI	30–36%	25–35%	64–70%	65–75%	17–34%	19–34%
Mobility: Moderate MSI	11–32%	14–30%	68–89%	70–86%	34–60%	34–62%

Abbreviations: rehab = rehabilitation; VI = vision impairment; HI = hearing impairment; MSI = musculoskeletal impairment. * Some participants were assessed to need both surgical/medical and rehab/APs interventions/services.

**Table 4 ijerph-19-04304-t004:** Proportion of participants assessed as having a clinical impairment who need * glasses, hearing aids, wheelchairs and protheses that were identified by WG some+ difficulty question.

Impairment Severity Level	Vision		Hearing	Mobility	
	Needs Distance Glasses	Needs Near Glasses	Needs Hearing Aids **	Needs Wheelchair	Needs UL/LL Prosthesis
	Some+/ Total Reported ^ N (%)	Some+/ Total Reported N (%)	Some+/ Total Reported N (%)	Some+/ Total Reported N (%)	Some+/ Total Reported N (%)
**Cameroon**					
Mild+	-	-	-	4/4	1/1
				100%	100%
Moderate+	10/17	-	26/36	4/4	1/1
	59%		72%	100%	100%
**Chile**					
Mild+	-	-	126/211	-	-
			60%		
Moderate+	-	-	60/78	-	-
			77%		
**India**					
Mild+	110/144	-	-	1/2	1/2
	76%			50%	50%
Moderate+	13/16	-	85/105	1/2	1/2
	81%		81%	50%	50%
**The Gambia**					
Mild+	315/529	1359/4002	-	-	-
	60%	34%			
Moderate+	260/423	-	-	-	-
	61%				
**Turkey**					
Mild+	-	-	-	9/9	2/2
				100%	100%
Moderate+	-	-	-	9/9	2/2
				100%	100%

Abbreviations: UL/LL = upper limb/lower limb; * “Need” includes both “unmet need” and “undermet need” for each assistive product; ^^^ Denominator includes participants who needed specific assistive products and who completed Washington Group questions in the respective functional domain; ** Hearing aid need includes all participants who needed a referral for audiological services and likely hearing aid need.

## Data Availability

The Gambia data presented in this study are still undergoing analysis and will be made openly available once complete. For the other datasets, we are unable to make the databases publicly available as we do not have participant consent for this. We can, however, share the databases with researchers upon request.

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
