# Peer review of "Exploring the Use of Washington Group Questions to Identify People with Clinical Impairments Who Need Services including Assistive Products: Results from Five Population-Based Surveys"

_ijerph, 2022, doi:10.3390/ijerph19074304_

Round 1

Reviewer 1 Report

This study aims to use secondary analysis of datasets from five population-based surveys across the functional domains of vision, hearing, mobility and cognition to address the sensitivity and specificity of the WG questions used for identification of people with clinically assessed impairments. It also aims to assess the proportion of people identified by the WG questions in need of surgical/medical and rehabilitation/AP services.

Lines 58-59: The authors point out in the introduction that access to rehabilitation and APs is limited. You should follow this with a sentence on what is needed to address the problem. Something that tells the reader that a better understanding of what is needed and where will add clarity to the aim of the study. That will link to the next paragraph explaining the measurement of functioning.

In Section 2.5 there is a change in font style. Please ensure all font is Times New Roman.

The authors make the important point in the Discussion that the WG should be used as an initial screening to indicate the need for further assessment.

The Discussion is well written and the recommendations are based on the findings of the study. The Strength s and Limitations are also discussed sufficiently. 

The Conclusions also are reasonable and do not attempt to overstate he findings. 

Reviewer 2 Report

The goal of this research is highly useful. The WG questions are becoming the international standard for identifying people with disabilities in surveys and censuses, and while they were not originally designed as a screening device, there has been much interest in using them as such. So a careful analysis of their efficacy in this regard is highly useful for countries designing programs to provide interventions for people with disabilities as efficiently possible.

The overall results suggest that the WG questions are a reasonable screening device. With the "a lot of difficulty" cutoff there is high specificity, but as the authors note, potentially many people who may need interventions will be missed. Using the "some difficulty" cutoff raises the selectivity to a reasonable level but still leaves a lot of room for improvement.

Of course the use of any screen is not that screen vs. perfection but versus the next best screen.  Therefore, it would've been more useful if a different screen was used as an alternative for comparison purposes. Is there an example of another commonly used screen? How do the WG questions measure up to that alternative? Clearly the WG questions as a screen is not perfect, but is it the best of what we have currently, or is there something that works better?  Still, because of the growing use of the WG questions, it is reasonable to test it on its own. However, I do think a bit more care needs to be used in presenting its strengths and weaknesses as I talk about below.

First, any such test relies on the "gold standard" (GS) upon which it is being measured. In this regard I have a few concerns.

A. The Mini-MSE as a GS for cognitive disabilities is suspect. This review suggests it should be retired:

Carnero-Pardo, C. "Should the mini-mental state examination be retired?." Neurología (English Edition) 29.8 (2014): 473-481.

And there are a number of papers showing other measures are better. For example here are three, but there are others:

Mancuso, Mauro, et al. "Using the Oxford cognitive screen to Detect cognitive impairment in stroke Patients: a comparison with the Mini-Mental state examination." Frontiers in Neurology 9 (2018): 101.

Van Patten, R., Britton, K., & Tremont, G. (2019). Comparing the Mini-Mental State Examination and the modified Mini-Mental State Examination in the detection of mild cognitive impairment in older adults. International psychogeriatrics31(5), 693-701.

Pinto, Tiago CC, et al. "Is the Montreal Cognitive Assessment (MoCA) screening superior to the Mini-Mental State Examination (MMSE) in the detection of mild cognitive impairment (MCI) and Alzheimer’s Disease (AD) in the elderly?." International psychogeriatrics 31.4 (2019): 491-504.

B. The RAM covers a broader range of mobility issues than the WG question on walking and climbing steps. Would things have been better if the WG Short Set Enhanced was used, which includes questions on upper body mobility? That possibility should be mentioned.

C. The vision GS picks up most vision difficulties, I would suppose, but there are other problems with vision, aren't there? like double vision, tunnel vision, and neurological vision impairments. When the WG vision question was tested in Vietnam versus two questions -- one on far vision and one on near vision -- the finding was the single question identified more people with vision problems, presumably because of these other vision difficulties. Is there any info on people who "passed" the GS but still needed interventions of some sort?

D. More fundamentally, the comparison in this paper is between binary GS (by domain) and an (admittedly limited) scaled response. What I'd like to know is how close were the people identified by the GS tests but not by the WG questions to the GS cutoff?  Did the WG miss people who were just over the GS line, or did they miss the folks who were way over that line.  If they mostly missed people who were just over the line, that increases my confidence in the WG as a screen to identify those in most need of services -- especially since countries may not have the resources to reach everyone and so would want to prioritize the ones in most need. (this also relates to my point 2 below in reporting results)

2. Results by domain

On line 244 the authors correctly report a sensitivity range of 44 to 85%. However that 44% is corresponding to those with mild hearing impairments. If we look by domain and degree we have higher sensitivity. We have what I would call a high sensitivity for Vision and pretty high for Mobility. And if we are only interested in moderate (+) than the WG works even better. I think the results by domain and level of disability should be highlighted a bit more.  Much screening for interventions is domain specific. If I have a good screen for vision and I am running a vision program, a relatively lower sensitivity for the WG hearing question wouldn't concern me. And some screening could be specifically for programs for the more seriously disabled, especially when there are budget constraints.  Of course it is still good to know who the WG are missing -- that is we may be missing people in need of services but who have more minor issues -- but for some purposes the quality of the WG as a screen may be better than for other purposes.

3. Table 3 is hard to understand

It took me a bit to figure out what was going on here. I think it would have got there quicker if the table heading had been: Proportion of participants to have a clinical impairment who need interventions who were identified as having a disability by the WG questions

This may seem like a picky thing, but it would've helped me.

4. In line 383, the authors suggest the RAM may be a better first stage screen. Is there any evidence for this? Clearly, it is worth testing other possibilities to get the best screen, but what is the reasoning behind suggesting this one?

5. In line 396, the authors suggest using the questions "without AT" but that is how the WG questions are used, so I am a little confused as to the point there.

6. In line 444, the authors report that respondents had problems differentiating "some" from "a lot".  That would also occur if the question had been "yes/no".  Also would've been problems if it had a been a scale like 1-5 or 1-10. Those thresholds are always tough and will arise with any self-reported screen -- but here we have the possibility of using the "some" as a screen whereas if we used yes/no we don't know how many "some's" would have answered "no."

Overall, I think this is a very useful paper. I would recommend publication if the issues I raised were addressed, and I don't think that would be difficult.
